# Universal symmetry-protected persistent spin textures in noncentrosymmetric crystals

Berkay Kilic [1], Sergio Alvarruiz[1], Evgenii Barts [1,2], Bertjan van Dijk[1], Paolo Barone [3] & Jagoda Sławińska [1] ✉

The significance of Mendeleev's periodic table extends beyond the classification of elements; it lies in its remarkable predictive power for discovering new elements and properties, revealing the underlying symmetrical patterns of nature that were only fully understood with the advent of quantum mechanics. Fundamental material properties, such as electron transport and magnetism, are also governed by crystal symmetry. In particular, spin transport depends on the spin polarization of electronic states, and recently discovered materials where the electron spin polarization is independent of momentum–a property known as persistent spin texture (PST)–promise extended spin lifetime and efficient spin accumulation. In this paper, we establish the general conditions for the existence of symmetry-protected PST in bulk crystals. By systematically analyzing all noncentrosymmetric crystallographic space groups, similar to elements in the periodic table, we demonstrate that PST is universally present in all nonmagnetic solids lacking inversion symmetry except those in the trivial space group P1. Using group theory, we identify the regions within the Brillouin zone that host PST and determine the corresponding directions of spin polarization. Our findings, supported by first-principles calculations of representative materials, open the route for discovering robust spintronic materials based on PST.

At the forefront of spintronics research lies the challenge of discovering and engineering materials that can efficiently generate and transport spin signals - an essential step toward realizing energy-efficient, spin-based electronics technologies[1]. Spin polarization is commonly introduced either through ferromagnetic materials, such as Fe-based compounds, or generated in nonmagnetic metals with strong spin–orbit coupling (SOC), such as Pt, via charge-to-spin conversion mechanisms[2,3]. However, stray magnetic fields from ferromagnets pose challenges in device applications, while strong SOC generally leads to rapid spin relaxation, limiting spin lifetimes[4–7]. In contrast, materials like graphene–characterized by weak SOC and high electron mobility–enable long-distance spin transport, but offer limited control over spin generation and manipulation[8]. Beyond these well-known cases, quantum materials with complex spin configurations of electrons offer more opportunities to explore. While they are interesting from the fundamental point of view, they can also reveal alternative relaxation mechanisms and open ways to achieve long spin lifetimes.

To make progress, it is essential to understand how crystal symmetries govern SOC in different materials. In solids, SOC appears as a relativistic, momentum-dependent effective magnetic field $\mathbf{B} \sim \nabla V(\mathbf{r}) \times \mathbf{p}$, where $\mathbf{p}$ is the electron momentum, and $V(\mathbf{r})$ is the crystal potential. This field interacts with the electron spin $\sigma$, causing

[1]Zernike Institute for Advanced Materials, University of Groningen, Nijenborgh 3, 9747AG Groningen, The Netherlands. [2]Quantum Materials Theory, Italian Institute of Technology, Via Morego 30, 16163 Genoa, Italy. [3]CNR-SPIN Institute for Superconducting and other Innovative Materials and Devices, Area della Ricerca di Tor Vergata, Via del Fosso del Cavaliere 100, I-00133 Rome, Italy. ✉e-mail: jagoda.slawinska@rug.nl

**a**

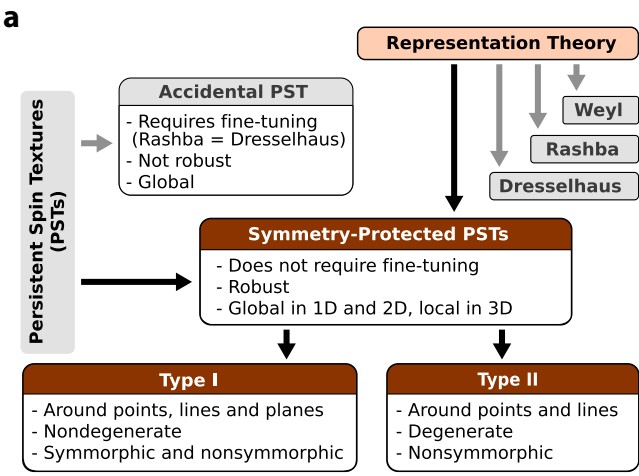

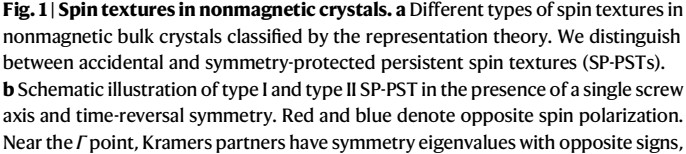

**b**

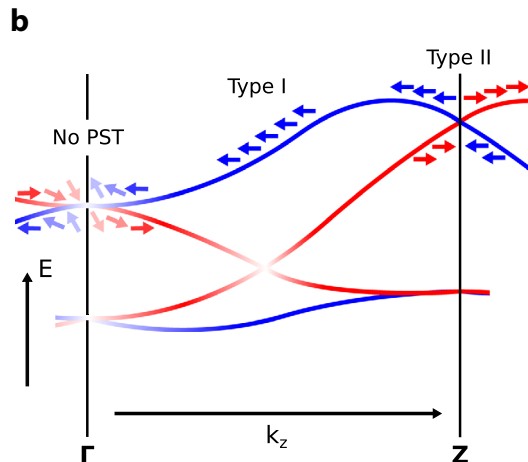

**Fig. 1 | Spin textures in nonmagnetic crystals. a** Different types of spin textures in nonmagnetic bulk crystals classified by the representation theory. We distinguish between accidental and symmetry-protected persistent spin textures (SP-PSTs). **b** Schematic illustration of type I and type II SP-PST in the presence of a single screw axis and time-reversal symmetry. Red and blue denote opposite spin polarization. Near the $\Gamma$ point, Kramers partners have symmetry eigenvalues with opposite signs, allowing nonzero spin components perpendicular to the high-symmetry direction, whereas at the $Z$ point, Kramers partners have eigenvalues with the same signs, preventing perpendicular spin components. Type I PST occurs around single (nondegenerate) bands along the high-symmetry lines, while type II PST emerges near the doubly-degenerate $Z$ point.

spin polarization of electronic states in **k**-space[9]. As a result, a variety of spin textures develop throughout the Brillouin zone (BZ)[10,11], playing a crucial role in spin transport properties. Spin textures are often categorized into three types based on their behavior around high-symmetry **k**-points: Rashba- featuring spin perpendicular to the momentum[12], Dresselhaus - a mix of parallel and perpendicular spin-momentum locking[13], and Weyl - parallel spin-momentum locking[14]. These types stem from effective **k** · **p** models, derived from the point group symmetries of a given **k**-point, but extending them beyond the linear-in-**k** approximation for the systems with general symmetry remains a challenge[15]. Moreover, particularly intriguing is the persistent spin texture (PST), where spin polarization remains unidirectional and independent of momentum[16], falling outside the conventional classification. This unique property minimizes spin dissipation and supports a long spin lifetime, which is highly desirable for spintronics devices[17,18], and calls for a more general framework that can comprehensively describe spin textures in materials with any crystal symmetries.

PSTs were originally discovered in two-dimensional (2D) semiconductor quantum wells, where the Rashba and Dresselhaus parameters were precisely balanced through adjustments in quantum well width, doping levels, and external electric fields[19]. However, PSTs in these systems are sensitive to perturbations, and the need for precise parameter tuning, combined with temperature limitations, impedes their practical applications. Recently, Tao and Tsymbal showed that symmetries in crystalline solids can enforce uniform spin polarization of states in specific regions of the BZ, creating symmetry-protected persistent spin textures (SP-PSTs)[20]. These SP-PSTs are robust against symmetry-preserving perturbations because they do not depend on the microscopic details of the crystal Hamiltonian. However, only a small class of crystals with nonsymmorphic symmetries has been shown to support PSTs around certain high-symmetry points in the BZ[20–27]. The general criteria for SP-PSTs have only been formulated for a subset of 2D materials[28]; beyond this subset, potential candidates are identified based on the presence of specific symmetries, which limit the range of available structures[29,30]. As a result, PST materials remain exceedingly rare.

In this Article, we use group theory analysis to establish the general conditions required for SP-PST. We discover that it is a universal property of all nonmagnetic crystals lacking inversion symmetry except those in the trivial space group P1. The crystal symmetries enforce PST around specific high-symmetry **k**-points, lines, or planes in the BZ. When bands near the Fermi level coincide with these regions, the material can support persistent spin helix states protected against perturbations by crystal symmetry. By identifying two types of SP-PSTs in degenerate and nondegenerate bands, we reveal the physical mechanisms behind their formation. In contrast to the previous studies that mostly relied on symmetry eigenvalue analysis for crystals with few symmetries, our approach is based on representation theory applied to all crystallographic space groups (SGs). Specifically, we use the irreducible corepresentations of double grey groups that take into account the full symmetry of crystals, encoding all the symmetry restrictions on the spin components of Bloch bands. Our approach is not only effective for crystals with many symmetries but can also be extended to predict all types of spin textures in nonmagnetic materials, which enables their systematic classification[10]. The applicability of our method is sketched in Fig. 1a, which also provides an overview of the different types of PST.

## Results and discussion
### Group theory analysis
The symmetry of the **k** point imposes constraints on electron spin polarization of Bloch states, $\psi_i$, following from the transformation rules of their spin matrix elements $\langle\sigma_\alpha\rangle_{ij} = \langle\psi_i|\sigma_\alpha|\psi_j\rangle$ given by[31]:

$$\langle\sigma'_\alpha\rangle_{ij} = \sum_{k,k'} D_{ik}(g)D^*_{jk'}(g)\langle\sigma_\alpha\rangle_{kk'}. \tag{1}$$

Here, the Pauli matrices $\sigma_\alpha$ (with $\alpha = x, y, z$) transform as an axial vector under the group element $g$: $\sigma'_\alpha = g\,\sigma_\alpha\,g^{-1}$. The matrix elements $D_{ij}(g) = \langle\psi_i|g|\psi_j\rangle$ define group representation on the Bloch states, and $i, j, k, k' = 1, \ldots, d$ where $d$ is the dimensionality of the representation at the **k** point. This method is the direct counterpart of the selection rules in quantum mechanics, commonly used in spectroscopy to identify possible transitions. They are allowed when the tensor product of the representations, under which the initial/final state and the perturbation potential transform, contains scalar representations in its Clebsch–Gordan decomposition[32].

While one could directly extend the approach of Ref. 31 by using representations of point groups that describe invariant regions in the

**Table 1 | Selection of the noncentrosymmetric crystallographic space groups with high-symmetry points, lines, and planes hosting SP-PST**

| Space group | | Points | | Lines | | Planes | | Example materials |
|---|---|---|---|---|---|---|---|---|
| *Monoclinic* | | | | | | | | |
| 4 | $P2_1$ | [010] | **Z, E, D, A** | [010] | $\Gamma Z$, YD, XA, EC | | | $Hf_{0.5}Zr_{0.5}O_2$[21] |
| 7 | *Pc* | [010] | **X, E, C, A** | | | [010] | $\Gamma YX$, ZDA | $BaAs_2$ |
| *Orthorhombic* | | | | | | | | |
| 19 | $P2_12_12_1$ | [100] | **X** | [100] | $\Gamma X$, **TR** | | | $Ag_2Se$[55] |
| | | [010] | **Y** | [010] | $\Gamma Y$, **UR** | | | |
| | | [001] | **Z** | [001] | $\Gamma Z$, **SR** | | | |
| 33 | $Pna2_1$ | [100] | **Y** | [100] | $\Gamma Y$, **UR** | [100] | $\Gamma ZY$, XUS | $BiInO_3$[20] |
| | | [010] | **X** | [010] | $\Gamma X$, **ZU** | [010] | $\Gamma ZX$, YTS | |
| | | [001] | **T** | | | | | |
| *Tetragonal* | | | | | | | | |
| 99 | $P4mm$ | | | [100] | $\Gamma X$, ZR | [100] | $\Gamma ZX$ | $BiFeO_3$[56] |
| | | | | [010] | XM, RA | [010] | XRM | |
| | | | | [1$\bar{1}$0] | $\Gamma M$, ZA | [1$\bar{1}$0] | $\Gamma ZM$ | |
| *Trigonal* | | | | | | | | |
| 154 | $P3_221$ | [001] | $\mathbf{\Gamma^*}$, $\mathbf{A^*}$ | [001] | $\Gamma A$, KH | | | Te[39] |
| | | | | [110] | $\Gamma K$, HA | | | |
| *Hexagonal* | | | | | | | | |
| 174 | $P\bar{6}$ | [001] | $\mathbf{\Gamma^*}$, K, H, $\mathbf{A^*}$ | [001] | $\Gamma M$, $\Gamma K$, $\Gamma A^*$, | [001] | $\Gamma MK$, LHA | $Ge_3Pb_5O_{11}$[57] |
| | | | | | LA, KH, HA | | | |
| 189 | $P\bar{6}2m$ | [001] | $\mathbf{\Gamma^*}$, $\mathbf{K^*}$, $\mathbf{H^*}$, $\mathbf{A^*}$ | [001] | $\Gamma M$, LA | [001] | $\Gamma MK$, LHA | $K_3Ta_3B_2O_{12}$[58] |
| | | | | [210] | ML | [1$\bar{1}$0] | $\Gamma KA$ | |
| *Cubic* | | | | | | | | |
| 198 | $P2_13$ | [010] | **X** | [010] | $\Gamma X$ | | | OsSi |
| | | | | [001] | **RM** | | | |
| | | | | [111] | $\Gamma R$ | | | |
| 216 | $F\bar{4}3m$ | [100] | W | [100] | XW | [1$\bar{1}$0] | $\Gamma LK$ | $Be_5Pt$ |
| | | [111] | $\mathbf{L^*}$ | [1$\bar{1}$0] | $\Gamma K$ | | | |

Note that we use the grey groups containing time-reversal symmetry (1'), which is omitted in the space group symbols for simplicity. Normal and bold fonts denote type I and type II PST, respectively. The spin direction is indicated in square brackets and expressed in terms of the conventional reciprocal lattice vectors. The definition of the reciprocal lattice vectors and the high-symmetry k-points follows the Setyawan–Curtarolo (SC) convention[59]. The symbol * denotes that there is a type I or type II PST in some representations, and no PST for the other representations. The last column lists materials examples known from the literature or identified in the present work. The full table is given in the Supplementary Note 4.

Brillouin zone, we instead use corepresentations proposed by Wigner to treat space group and time reversal operations on equal footing[32]. This approach allows us to classify unidirectional spin textures in any nonmagnetic material, even in nontrivial cases where the **k** point is invariant under a point symmetry transformation combined with time reversal. To this end, we employ the irreducible corepresentations (irreps) of double magnetic space groups of type II, called grey groups[33,34], taken from the Bilbao Crystallographic Server[35,36]. Although it may seem counterintuitive to use magnetic groups to study non-magnetic materials, grey groups appear particularly suitable because they explicitly contain time-reversal symmetry as a group element. This guarantees that time-reversal symmetry is unbroken, as is the case in nonmagnetic materials. To correctly describe half-integer spin states under spatial symmetries, we employ double group representations. Our framework, based on the irreducible representations of double grey groups, enables a systematic and unified treatment of all symmetries in nonmagnetic crystals within the formalism of representation theory.

In order to identify PST regions in all nonmagnetic non-centrosymmetric crystals, we apply Eq. (1) to all symmetry elements of the little group for a given irrep and identify **k** points where only a single spin component is allowed. Our results for all noncentrosymmetric space groups are presented in Supplementary Table S1. In addition, Table 1 provides a condensed version containing a selection of representative space groups. For each symmetry group, the table lists the regions in the BZ−including high-symmetry points, lines, and planes−where SP-PST is present, the direction of the allowed spin polarization, as well as example materials.

Surprisingly, our analysis reveals that all 138 noncentrosymmetric space groups, except for the trivial group P1, have regions where PST occurs, indicating that this property is universal among nonmagnetic bulk crystals without inversion symmetry.

We further categorize PSTs into two types: type I PST, which corresponds to a nondegenerate band, and type II PST−to degenerate bands. For type I PST, Eq. (1) reduces to $\langle\sigma'_\alpha\rangle = \langle\sigma_\alpha\rangle$. For example, if $g$ is a twofold rotation around the $z$ axis, this equation becomes $(-\langle\sigma_x\rangle, -\langle\sigma_y\rangle, \langle\sigma_z\rangle) = (\langle\sigma_x\rangle, \langle\sigma_y\rangle, \langle\sigma_z\rangle)$, enforcing PST with spin polarization parallel to the rotational axis, i.e., $\langle\sigma_{x,y}\rangle = 0$ and $\langle\sigma_z\rangle \neq 0$ (see Fig. 1b). Similarly, mirror symmetry enforces PST with spin polarization perpendicular to the mirror plane, and the presence of symmetries with different polarization axes constrains $\langle\sigma_{x,y,z}\rangle = 0$. Nondegenerate bands can occur only at **k** points that are not invariant under any antiunitary symmetry squaring to −1, such as time-reversal symmetry alone or in combination with crystal symmetries, which excludes many points, such as $\Gamma$, $X$, $Y$, and $Z$ in primitive lattices, from supporting type I PST. We find that type I PSTs can occur around high-symmetry points only in non-primitive and trigonal primitive lattices, where such antiunitary symmetries can be broken at high-symmetry points.

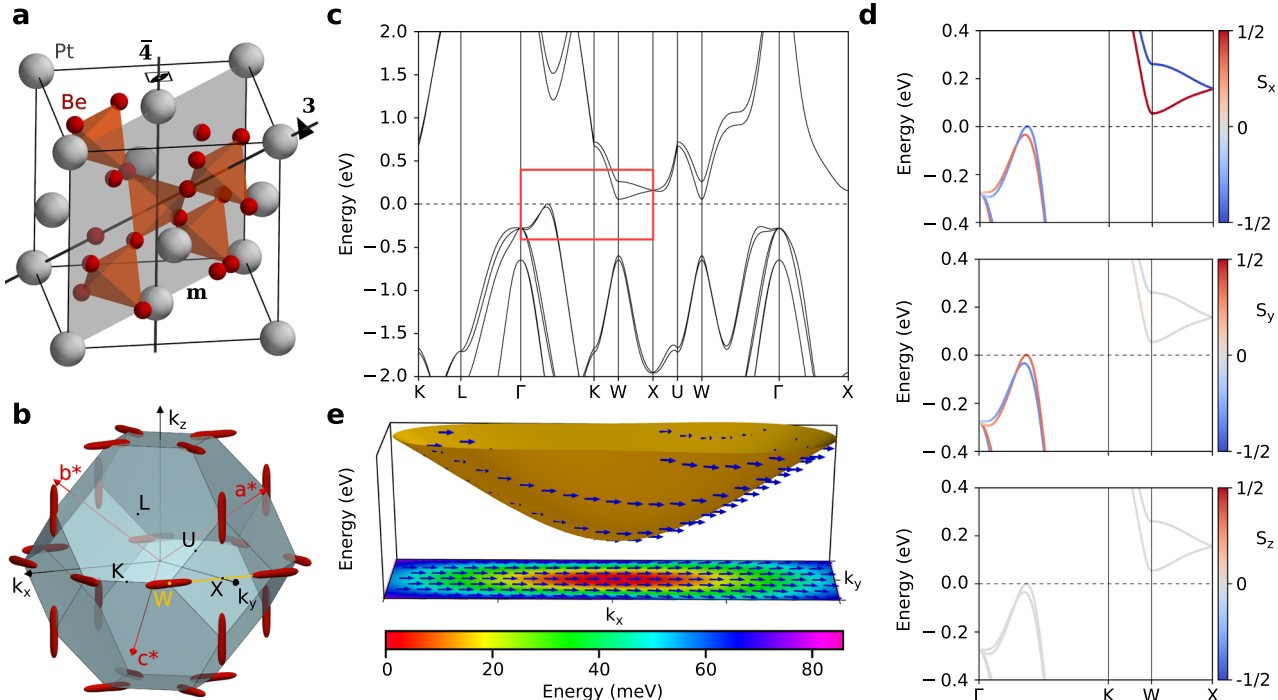

**Fig. 2 | Electronic properties and persistent spin texture in Be₅Pt, an example of type I PST. a** Crystal structure and symmetries: 4-fold rotoinversion axes parallel to the conventional lattice vectors, 3-fold rotation axes parallel to the body diagonals, and mirror planes perpendicular to the face diagonals of the cube. **b** Brillouin zone (BZ) with high-symmetry *k*-points and pockets of constant energy $E = 50$ meV above the conduction band minimum (CBM). **c** Electronic band structure calculated along high-symmetry lines. The region marked by the red rectangle is magnified in panel (**d**), which additionally presents *x, y*, and *z* components of the spin texture. **e** Lowest conduction band around the *WX* line with arrows representing the spin texture aligned along the *x*-axis. The color map shows the energy measured from the CBM. BZ and high-symmetry points follow the convention of Ref. [59].

Famous material realizations of type I PST are transition metal dichalcogenides, such as MoS₂ and WS₂, where the *K* point in their honeycomb BZ displays robust out-of-plane PST, extending over a large region away from the *K* point[37,38]. Type I PST also occurs in bulk tellurium crystals along the vertical line connecting the *K* and *H* points, where the three-fold symmetry protects the PST[39,40]. Near these high symmetry points, spin canting arises from the admixture of other bands with different spin polarization. These bands can be included perturbatively, following the Luttinger-Kohn model, and therefore in low-energy **k · p** expansion, stronger band separation in energy leads to persistent uniaxial spin polarization over a larger region in reciprocal space.

In type II PST, which involves degenerate bands, the analysis is more complicated. In general, even if uniaxial spin polarization is enforced in two degenerate bands, spin transfer in the vicinity of the band crossing can curl spins into a vortex-like spin texture, such as Rashba (or Dresselhaus). To avoid it in type II PST, the conditions to prevent spin flips or canting are that the irrep includes only one identity matrix $\pm \mathbb{I}_{d \times d}$ corresponding to a crystal symmetry different from the identity element of the group. This condition implies that the states forming the degeneracy must have the same eigenvalues for one of the symmetry elements in the little group. For instance, at the $\Gamma$ point in SG 4 ($P2_1$), the matrix of the two-fold screw rotation is

$$\Gamma(\widetilde{C}_{2z}) = \begin{pmatrix} \pm i & 0 \\ 0 & \mp i \end{pmatrix}, \tag{2}$$

and by applying Eq. (1), we find $\langle \sigma_{x,y} \rangle_{11,22} = 0$, $\langle \sigma_z \rangle_{11,22} \neq 0$, $\langle \sigma_{x,y} \rangle_{12,21} \neq 0$ and $\langle \sigma_z \rangle_{12,21} = 0$. These conditions lead to nonzero spin components in the *x, y*, and *z* directions near the $\Gamma$ point because, as one moves away from this high-symmetry point, the bands split and their wavefunctions become linear combinations of the unperturbed basis states, e.g., $a_k |\psi_1\rangle + b_k |\psi_2\rangle$, acquiring non-zero in-plane spin components

proportional to $\langle \sigma_{x,y} \rangle_{12,21}$. In contrast, at the *Z* point, the matrix is

$$Z(\widetilde{C}_{2z}) = \begin{pmatrix} \pm 1 & 0 \\ 0 & \pm 1 \end{pmatrix}, \tag{3}$$

which results in $\langle \sigma_{x,y} \rangle_{ij} = 0$ and $\langle \sigma_z \rangle_{ij} \neq 0$ for all *i, j*, enforcing type II PST with spin polarization parallel or antiparallel to the screw rotation axis (see Fig. 1b).

The need for such a diagonal symmetry representation to preserve uniaxial spin polarization near a high-symmetry region can be understood intuitively via the **k · p** expansion. Full derivations of the **k · p** models by applying Eq. (1) to the velocity are provided in Supplementary Note 1, but the key idea is as follows. The general SOC Hamiltonian takes the form $H_{SOC} = \sum_{\alpha\beta} g_{\alpha\beta} k_\alpha \sigma_\beta$, where $g_{\alpha\beta}$ ($\alpha = x, y, z$) are coupling constants constrained by the crystal symmetry. When projected onto the subspace of degenerate bands, the spin operator $\sigma_\beta$ is replaced by $\sum_{ij} \langle \sigma_\beta \rangle_{ij} |\psi_i\rangle \langle \psi_j|$. If the symmetry representation is diagonal, as is the case of the *Z* point, the matrix elements of the in-plane spin operators $\langle \sigma_\beta \rangle_{ij}$ vanish for $\beta = x, y$ and all *i, j*. As a result, SOC effectively reduces to $H_{SOC} = \sum_\alpha g_{\alpha z} k_\alpha \sigma_z$, since $\sigma_{x,y}$ act trivially within this subspace. Consequently, the spin polarization remains unidirectional in the vicinity of the high-symmetry point.

Symmetries with irreps of the form in Eq. (3) enforce PST along their invariant axes. Notably, such irreps can only occur in the presence of nonsymmorphic symmetries, where the fractional translation part of the symmetry operator allows for real eigenvalues at the BZ edges, permitting the diagonal matrices as in Eq. (3). PST of this type was first proposed through an eigenvalue analysis of glide mirror symmetries in orthorhombic space groups[20]. However, our approach provides a systematic generalization applicable to all possible symmetries and space groups. Besides, we discover that in orthorhombic crystals, there are more points and lines around which PST is symmetry-

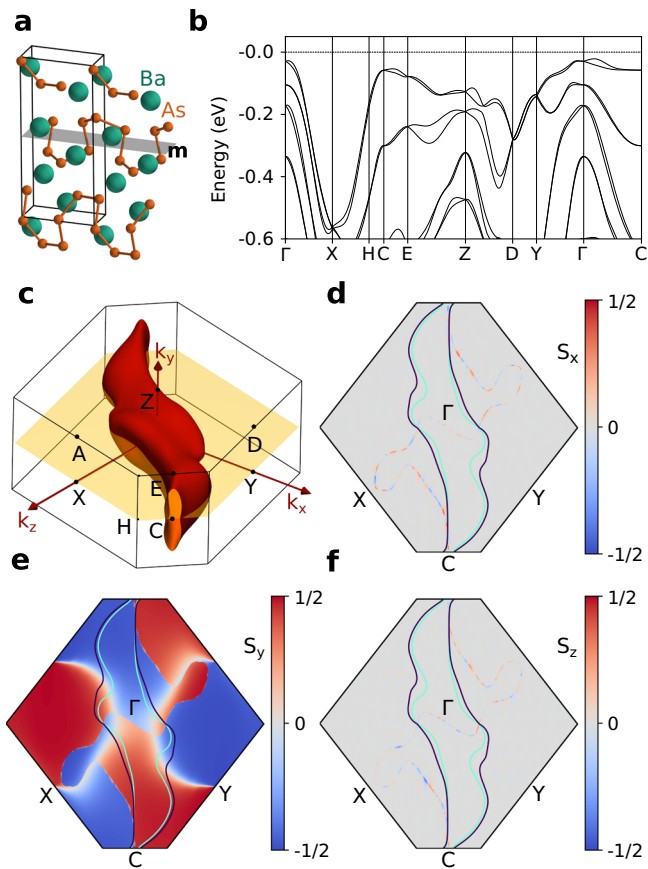

**Fig. 3 | Geometry and electronic properties of BaAs₂, an example of type I PST.** **a** Crystal structure with the mirror plane marked in grey. **b** Valence band structure along the high-symmetry path; the band gap is 0.34 eV. **c** Constant energy surface at $E = -50$ meV. **d–f** Spin texture of the topmost valence band at the $\Gamma YX$ plane, marked as yellow in panel (**c**). The constant energy contours correspond to $E = -50$ meV.

enforced, particularly along the edges of glide mirror planes (see Table S1).

## First-principles calculations

Table 1 summarizes several materials with SP-PST previously reported in the literature, all of which are consistent with our symmetry-based predictions. To further show the predictive power of our approach, we identified additional representative materials that host different types of SP-PST. The initial search was conducted using the AFLOW materials database[41], which, despite containing nonrelativistic electronic structures, provided a useful first indication of whether PST-associated paths in the BZ have bands near the Fermi level. To guide our search, we selected materials with strong SOC to increase band splitting and small unit cells to streamline the calculations. We then performed detailed density functional theory (DFT) calculations on a few promising candidates, with the computational details for each material described in the "Methods" section.

First, we focus on a cubic crystal Be₅Pt (SG 216) with type I SP-PST around a line. The material was previously synthesized; it has a large SOC and a small unit cell consisting of six atoms[42]. Its crystal structure and electronic properties are shown in Fig. 2, and they agree well with the previous studies[43]. The conduction band minimum (CBM), located at the $W$ point, shows a huge spin-splitting of 205 meV, enabling applications well above room temperature. According to Table 1, Be₅Pt should have SP-PST of type I around the $W$ point and along the $XW$ line, with spins parallel to the $x$-direction in reciprocal space. This finding is fully confirmed by our spin-resolved band structures plotted in Fig. 2d.

Additionally, we explored the spin texture across the entire electron pockets near the CBM. For energy $E = 50$ meV, the pockets take on a cigar-like shape, growing larger at higher energies, though the spin texture remains well-aligned along the $x$ direction, with minor deviations in larger pockets (see Fig. 2e). Last, we estimated the spin Hall angle, finding it comparable to Pt (5–10%, see Fig. S1 in SM). The combination of large spin-splitting, robust PST across the pockets, and a large spin Hall angle makes Be₅Pt promising for spintronics devices.

Figure 3 shows the calculated properties of BaAs₂, which crystallizes in a monoclinic lattice (SG 7). The structure consists of chiral chains of As separated by Ba cations. The only symmetry of the material is a mirror plane perpendicular to the $y$-axis. This symmetry implies type I SP-PST in the $\Gamma YX$ and $ZDA$ planes (see Table 1). Since BaAs₂ is a semiconductor with the VBM in the $\Gamma YX$ plane, it is convenient to analyze the spin texture of the valence bands. Near the Fermi level, the constant energy surface consists of two sheets extending over the entire Brillouin zone along the $\Gamma YX$ plane; one of them is visualized in Fig. 3c. The spin texture of the topmost valence band is shown in Fig. 3d–f, demonstrating SP-PST in the $y$ direction. Consistent with group theory predictions, the spin texture is unidirectional across the entire plane, except in a few regions that appear white in Fig. 3e, where the $S_x$ and $S_z$ components are also present. These regions correspond to the crossing with the second valence band. Because the $\Gamma YX$ plane hosts a type I PST, the band degeneracies disrupt PST.

Finally, we calculated the properties of the chiral cubic crystal OsSi (SG 198), which belongs to the B20 materials class. Figure 4 shows its structural, electronic and spin properties. Both VBM and CBM lie in the $\Gamma$–$X$ line (see Fig. 4b), and the bands near the Fermi level form several tiny pockets, as illustrated in Fig. 4c. The pockets close to VBM, shown in Fig. 4d, e, feature type I PST along the $y$ axis. Moreover, the CBM is located near the X point, which exhibits type II PST also along the $y$ axis. The type II PST is enforced by the 2-fold screw rotation, which plays a similar role to the glide-mirror symmetries studied in Ref. 20. Consequently, the band degeneracies do not disrupt the PST around this point, as shown in Fig. 4e. We believe that SP-PST will be crucial for sustaining long-range transport of spin density generated via the Rashba–Edelstein effect recently studied in this material[40].

In summary, we demonstrated that persistent spin textures can be found in noncentrosymmetric materials with any crystal symmetries, contrary to the common belief that they are an exotic property limited to a few special material classes. The exact knowledge of the PST location in the BZ for each space group allows for an easy design of materials with PST near the Fermi level, where they significantly influence the material properties and play a crucial role in spin generation and spin dynamics. Additionally, our theoretical approach based on representation theory not only allows for the classification of all types of spin textures in nonmagnetic solids but can also be extended to broader material classes, including altermagnets and other magnetic systems, as well as centrosymmetric structures with hidden spin textures[44,45]. We believe that our findings represent a significant step toward identifying materials that combine strong spin-orbit coupling with extended spin lifetimes, addressing one of the most critical challenges in spintronics.

## Methods

We performed first-principles calculations for all the materials using the Quantum Espresso package[46,47]. The ion-electron interaction was treated using the fully relativistic projected-augmented wave pseudopotentials from the pslibrary database[48], and the electron wave functions were expanded in a plane-wave basis using converged kinetic energy cutoff values. The exchange and correlation interaction was taken into account via the generalized gradient approximation (GGA) parameterized by the Perdew, Burke, and Ernzerhof (PBE) functional[49]. The atomic coordinates of the structures were relaxed

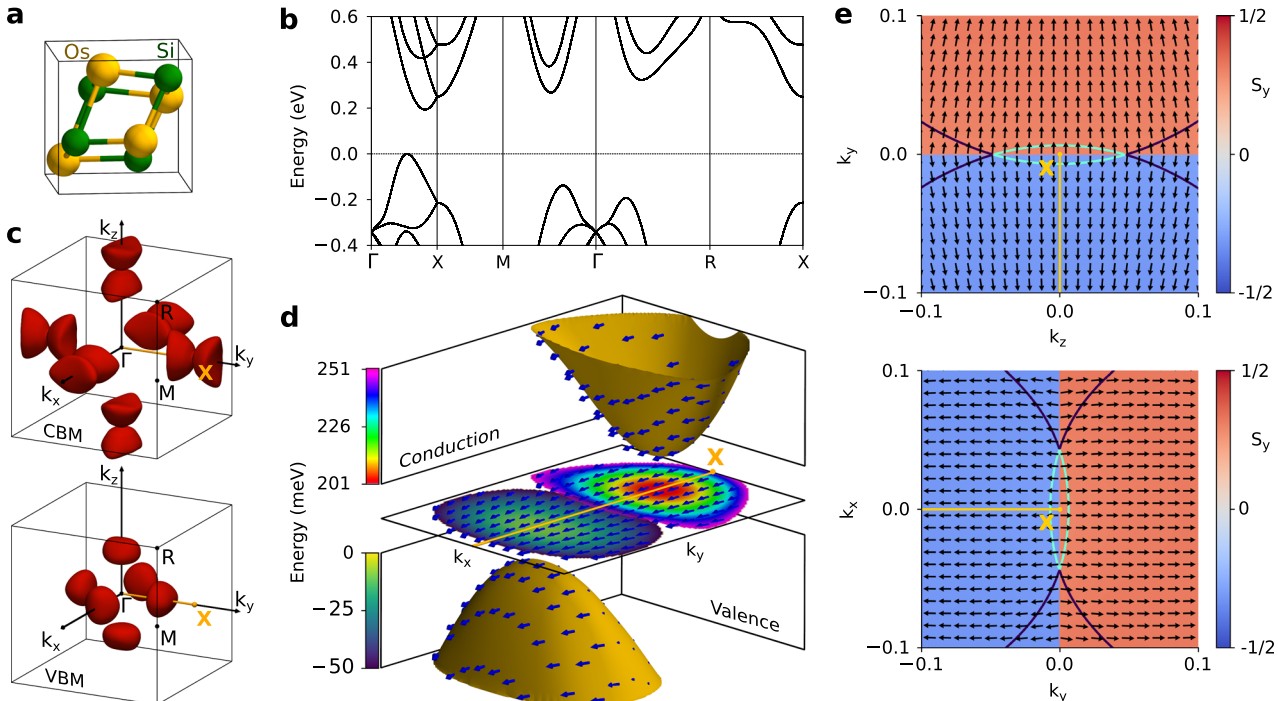

**Fig. 4 | Chiral OsSi as an example of SP-PST of type II. a** Cubic unit cell. **b** Electronic structure along the high-symmetry lines; the energy gap is 200 meV. **c** Constant energy surfaces of the lowest (highest) conduction (valence) bands. The energy levels are set to +50 meV and −50 meV with respect to CBM and VBM, respectively. **d** Three-dimensional view of the valence and conduction bands closest to the Fermi level. The arrows represent the spin texture. The central color maps display the energy profiles. **e** Maps of the spin texture at the *xy* and *yz* planes. The wave vector is given with respect to the *X* point, which is set to zero. The constant energy contours correspond to *E* = 50 meV.

with the convergence criteria for energy and forces to $10^{-8}$ Ry and $10^{-4}$ Ry/bohr, respectively. The electronic convergence threshold was set to $10^{-9}$ Ry. The BZ sampling at the DFT level was performed following the Monkhorst–Pack scheme[50], using *k*-meshes converged separately for each material. SOC was taken into account self-consistently in all the calculations except for the structural optimizations. The post-processing calculations of spin textures and spin Hall conductivity were performed using the PAOFLOW code[51,52].

Be₅Pt (SG 216) was simulated in a cubic unit cell using the experimental lattice constant $a = 5.97$ Å[42]. We used the fixed occupations scheme and the *k*-points grids of $48 \times 48 \times 48$. The kinetic energy cutoff for wavefunctions was set to 280 Ry. BaAs₂ was calculated with a monoclinic cell with experimental lattice parameters $a = 6.55$ Å, $b = 12.53$ Å, $c = 8.04$ Å, $\beta = 127.75$[53]. We used Gaussian smearing of 0.001 Ry and the *k*-point grids of $12 \times 12 \times 12$. The kinetic energy cutoff for wavefunctions was set to 110 Ry. OsSi was modeled in a cubic unit cell with the experimental lattice constant $a = 4.73$ Å[54]. We used the fixed occupations scheme and the *k*-points grids of $20 \times 20 \times 20$. The kinetic energy cutoff for wavefunctions was set to 110 Ry.

## Data availability

The data associated with the manuscript is available at DataverseNL https://doi.org/10.34894/N4LUQ8.

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

## Acknowledgements

J.S. acknowledges the Rosalind Franklin Fellowship from the University of Groningen. We acknowledge the Dutch Research Council (NWO) for the grants NWA.1418.22.014 and OCENW.M.22.063. The calculations were carried out on the Dutch national e-infrastructure with the support of SURF Cooperative (EINF-8924) and on the Hábrók high-performance computing cluster of the University of Groningen.

## Author contributions

B.K. conceived and performed the irrep-based symmetry analysis and formulated a search algorithm. S.A. performed first-principles calcula-tions and prepared the figures. P.B. suggested the use of double grey groups and established with E.B. and B.K. the link with k · p theory. B.v.D. automated the extraction of representation matrices and implemented the algorithm. All the authors contributed to the writing, discussions and data analysis. J.S. conceived and supervised the project.

## Competing interests

The authors declare no competing interests.
