## [Peer Review File · Nature Communications]

Universal symmetry-protected persistent spin textures in noncentrosymmetric crystals

Corresponding Author: Professor Jagoda Sławińska

Version 0:

Reviewer comments:

Reviewer #1

(Remarks to the Author)

In this work, based on group theoretical analysis, the authors thoroughly investigated the symmetry dictated spin polarization in all noncentrosymmetric SGs. In particular, they identify the regions in the Brillouin zone hosting the PST, which are exemplified in realistic materials based on DFT calculations. Interestingly, it is found that the PST is universally present in all nonmagnetic and noncentrosymmetric solids. The results undeniably broaden the range of the materials sustaining PST.

However, the authors should address the following major points before the manuscript may be considered for publication:

- (1) In the section of group theory analysis, the authors employ the irreps of double magnetic space groups of type II to identify the k point hosting PST. This should be elaborated. Why is the double magnetic space group of type II equivalent to nonmagnetic space group? For a given symmetry operation and SG, how to calculate the representation matrix (e.g., Eqs. 2 and 3)? I suggest the authors add derivation and analysis details for a specific example in supplementary material.
- (2) Following comment 1, previous work demonstrated that the little cogroup at the Brillouin zone border point should be extended due to the inclusion of the translation operation for nonsymmorphic space groups [e.g., PRB 94, 155124 (2016); PRB 110, L121125 (2024)]. Thus, one cannot analyze the SOC Hamiltonian based on point group symmetry analysis for the high symmetry k points at the Brillouin zone border. The authors should comment on this and discuss the possible accordance between those different approaches.
- (3) In Table S1, the authors only list the spin direction at high-symmetry points and lines. However, the spin texture spans large area around points or lines in the Brillouin zone. The authors should work out the k.p Hamiltonian around high-symmetry points or lines to clearly show the deviation from idea PST. While I understand it is a heavy task to work out for all SGs, the authors should provide several representative examples.
- (4) Since the authors focus on the noncentrosymmetric space groups, I think the title should be more specific by adding noncentrosymmetric somewhere. Then, the centrosymmetric space groups listed in Table S1 should be removed.

Reviewer #2

(Remarks to the Author)

In this theoretical work the authors establish the general conditions for the existence of symmetry-protected persistent spin textures in nonmagnetic solids using group theory. They systematically analyze all 230 crystallographic space groups and prove that persistent spin textures are universally present in all nonmagnetic solids lacking inversion symmetry. They also determine the persistent spin textures location in the Brillouin Zone for each space group and identify two types of symmetry-protected persistent spin textures in degenerate and non-degenerate bands. A few example are discussed in detailed with first-principles calculations supporting the theory group classification.

The work is sound. The author clearly discuss the scopes of their research, the methodology and the results obtained. I found the paper nicely written and the results extremely timing and relevant.

In particular I found the following aspects particularly relevant:

- The analysis performed here nicely organized the results of past investigations based on the analysis of few symmetry and systematically address all the crystallographic space groups providing new knowledge in the spin properties of a large class of materials.
- The approach here discussed can be generalized and extended to the analysis of a broader class of materials of high interest, including altermagnets.
- The precise localization of the persistent spin textures for each space group will allow in principle to layout a protocol for

designing materials with persistent spin textures near the Fermi level.

- Finally a remarkable outcome of the research is to show that persistent spin textures are not exotic property limited to a few special material classes, but actually are universally present in all nonmagnetic solids lacking inversion symmetry.

I do not have specific criticisms, just a couple of comments.

- It is a bit weird to find the main result on the investigation in the supplementary material, however I understand the need of put table I in there. I only recommend the authors to emphasize more the presence of the full analysis in the supplementary material Table I and to make reference to the table when discussing the symmetry-based results.

- I think that the introduction would benefit from a wider introduction of the topic. Although clear to specialists, I think that explaining more the relevance of persistent spin texture and the implication for material properties would help the audience of material scientists to better appreciate the relevance of the results.

In conclusion, I have minor suggestions to improve the clarity of the manuscript and I recommend the manuscript for publication after minor revision from the authors.

Reviewer #3

(Remarks to the Author)

This study presents a new theoretical framework for identifying symmetry-protected persistent spin textures (SP-PSTs) in nonmagnetic solids. By systematically analyzing all 230 crystallographic space groups (SGs) using irreducible corepresentations of double grey groups, the authors demonstrate that SP-PSTs are a near-universal property of noncentrosymmetric crystals. The work classifies SP-PSTs into two types (degenerate and nondegenerate bands), identifies their locations in the Brillouin zone (BZ), and validates predictions via first-principles calculations on representative materials (e.g., Be_5Pt , Ag_2Se). The findings challenge the conventional view that PSTs are exotic phenomena limited to specific material classes, offering a transformative approach for designing spintronic materials with robust spin lifetimes. I suggest to publish this paper in NC after the authors address the following comments:

(1) Ambiguity in Universality Claims

The abstract states that PSTs are "universally present in all nonmagnetic solids lacking inversion symmetry." However, the main text clarifies that PSTs are absent in the trivial space group $P1$ (see Sec. 4 and Supplementary Table S1). This inconsistency risks overstating the universality of PSTs.

Recommendation: Revise the abstract to explicitly exclude SG $P1$, e.g.,

"We demonstrate that PSTs are universally present in all noncentrosymmetric nonmagnetic crystals except those in the trivial space group $P1$."

(2) Rationale for Using Grey Groups Over Double Groups

The authors employ irreducible corepresentations of double magnetic space groups of type II (grey groups) rather than conventional double groups. While this choice enables the classification of SP-PSTs under time-reversal symmetry (TRS), the rationale remains unclear.

Why do grey groups (which unify space group operations and TRS) provide a more robust framework for identifying type-II PSTs (e.g., in degenerate bands) compared to double groups (which focus on spin-1/2 states without TRS)?

Does the exclusion of TRS in double groups inherently fail to capture type-II PSTs? If so, explain how grey groups resolve this limitation (e.g., via eigenvalue constraints in Eq. 3).

Recommendation: Expand the Methods section to justify the grey group framework, emphasizing its necessity for predicting PSTs in systems with nonsymmorphic symmetries (e.g., glide mirrors) or TRS-enforced degeneracies.

Version 1:

Reviewer comments:

Reviewer #1

(Remarks to the Author)

I have carefully studied the authors' response letter and revised manuscript, which have addressed my concerns. I would like to recommend the revision for publication.

Reviewer #3

(Remarks to the Author)

This paper can now be published.

Groningen, June 5th, 2025

Response letter for the manuscript NCOMMS-24-72286-T

Response to Reviewer 1

In this work, based on group theoretical analysis, the authors thoroughly investigated the symmetry dictated spin polarization in all noncentrosymmetric SGs. In particular, they identify the regions in the Brillouin zone hosting the PST, which are exemplified in realistic materials based on DFT calculations. Interestingly, it is found that the PST is universally present in all nonmagnetic and noncentrosymmetric solids. The results undeniably broaden the range of the materials sustaining PST. However, the authors should address the following major points before the manuscript may be considered for publication:

We greatly appreciate the positive assessment of our manuscript.

(1) In the section of group theory analysis, the authors employ the irreps of double magnetic space groups of type II to identify the k point hosting PST. This should be elaborated. Why is the double magnetic space group of type II equivalent to nonmagnetic space group? For a given symmetry operation and SG, how to calculate the representation matrix (e.g., Eqs. 2 and 3)? I suggest the authors add derivation and analysis details for a specific example in supplementary material.

We agree that the justification for using the type-II double magnetic space (grey) groups was not adequately explained in the main text. In nonmagnetic materials, time-reversal symmetry (T) is an inherent symmetry of the system. A possible approach is then to treat separately spatial symmetries (using space groups G , accounting for both point-group and translation symmetry elements) and T, requiring to keep track of the commutation relations with the crystal symmetry

elements, as well as treating all anti-unitary symmetry elements of the form gT , where g is a crystal symmetry belonging to the space group G . While the approach is usually applied to specific problems, we believe that it is not the most convenient one for a systematic analysis of all possible nonmagnetic crystals, since the calculation can get very complicated for high-symmetry groups such as hexagonal and cubic space groups.

An equally valid and more effective approach is instead to treat simultaneously and at the same level of group/representation theory all symmetries, comprising both unitary and anti-unitary elements and allowing for a systematic classification using irreducible (co-)representations. Formally, this implies the inclusion of T as an element of the group, leading to a magnetic grey group $G \otimes T$ where the number of unitary and anti-unitary elements g and gT must be the same to enforce the closure property of the (magnetic) group. Representation theory for magnetic space groups is a well-established one (see, e.g., Bradley and Cracknell, *The mathematical theory of symmetry in solids*, Clarendon Press - Oxford 1972, chapter 7), with tables of irreducible (co)representations available in textbooks (Miller and Love, *Tables of Irreducible Representations of Space Groups and Co-Representations of Magnetic Space Groups*, Pruett Press, Denver 1967) or in online databases such as Bilbao Crystallographic Server (BCS, <https://www.cryst.ehu.es/>) for all groups (including nonsymmorphic ones) and wave-vectors (including Brillouin zone borders).

Similarly, we adopted double groups as they allow one to explicitly take into account the symmetry properties of half-integer spins, which are essential when the Hamiltonian describing the system contains terms that explicitly depend on electronic spins, as the spin-orbit coupling. We have described in detail the motivation behind our choice in the manuscript, and cited the relevant references, where the form and/or derivation of representation matrices are provided.

(2) Following comment 1, previous work demonstrated that the little cogroup at the Brillouin zone border point should be extended due to the inclusion of the translation operation for nonsymmorphic space groups [e.g., PRB 94, 155124 (2016); PRB 110, L121125 (2024)]. Thus, one cannot analyze the SOC Hamiltonian based on point group symmetry analysis for the high symmetry k points at the Brillouin zone border. The authors should comment on this and discuss the possible accordance between those different approaches.

The Reviewer is right when she/he says that the little group of wave-vector k in nonsymmorphic groups need to include (fractional) translation operations entering in screw/glide elements. However, it is a well-established result of group theory that, since the lattice translations always form a normal self-conjugate subgroup of any space group, the little group of any wave-vector must be isomorphic to a point group, implying that point-group symmetry analysis is also appropriate for dealing with rotational aspects of non-symmorphic space groups. While for symmorphic space groups any little group is always one of the 32 crystallographic point groups,

for nonsymmorphic groups one needs to keep track of phase factors involving fractional translations, as discussed in textbooks such as Bradley and Cracknell (*The mathematical theory of symmetry in solids*, Clarendon Press - Oxford 1972, chapter 4) or Dresselhaus, Dresselhaus, Iorio (*Group theory application to the Physics of Condensed Matter*, Springer-Verlag Berlin Heidelberg 2008, chapter 10). As mentioned in our reply to the previous comment of Reviewer 1, tables for the little group of any wave-vector for each (magnetic) space group are available in textbooks (e.g., Miller and Love, *Tables of Irreducible Representations of Space Groups and Co-Representations of Magnetic Space Groups*, Pruett Press, Denver 1967) or online databases (BCS, <https://www.cryst.ehu.es/>). Specifically, we used tables from the Bilbao Crystallographic Server for our systematic analysis in this work.

(3) In Table S1, the authors only list the spin direction at high-symmetry points and lines. However, the spin texture spans large area around points or lines in the Brillouin zone. The authors should work out the k.p Hamiltonian around high-symmetry points or lines to clearly show the deviation from idea PST. While I understand it is a heavy task to work out for all SGs, the authors should provide several representative examples.

It is true that the persistent spin textures span a large area around the points, lines, and planes listed in Table S1. The size of the region will, in principle, depend on the material. In general,

- 1) The spin texture around a given (high-symmetry) point \mathbf{k}_0 in the Brillouin zone can always be deduced by knowing the allowed spin-matrix elements at the targeted \mathbf{k}_0 , listed in Table S1.
- 2) If all irreducible representations at the targeted point only allow for a uniaxial component, no spin-mixing affecting the persistent uniaxial spin polarization occurs.
- 3) As a rule of thumb, one can expect that for a band with uniaxial spin polarization at high-symmetry points/lines, the stronger the energy separation from other bands, the larger the region is in reciprocal space where persistent uniaxial spin polarization is realized.

Following the reviewer's suggestions, we have added a supplementary section, where we derive the k.p Hamiltonian for a representative example of SG 4, which is also considered in the manuscript. The general formulation and the derivation of k.p Hamiltonians for all SGs is beyond the scope of the current work, and it will be provided in a separate study in the future.

(4) Since the authors focus on the noncentrosymmetric space groups, I think the title should be more specific by adding noncentrosymmetric somewhere. Then, the centrosymmetric space groups listed in Table S1 should be removed.

We agree with the reviewer and emphasized that our conclusion is valid for noncentrosymmetric space groups only. We have also removed the centrosymmetric groups from the Table.

Response to Reviewer 2

In this theoretical work the authors establish the general conditions for the existence of symmetry-protected persistent spin textures in nonmagnetic solids using group theory. They systematically analyze all 230 crystallographic space groups and prove that persistent spin textures are universally present in all nonmagnetic solids lacking inversion symmetry. They also determine the persistent spin textures location in the Brillouin Zone for each space group and identify two types of symmetry-protected persistent spin textures in degenerate and non-degenerate bands. A few examples are discussed in detail with first-principles calculations supporting the theory group classification.

The work is sound. The author clearly discusses the scopes of their research, the methodology and the results obtained. I found the paper nicely written and the results extremely timely and relevant. In particular I found the following aspects particularly relevant:

- The analysis performed here nicely organizes the results of past investigations based on the analysis of few symmetries and systematically addresses all the crystallographic space groups providing new knowledge in the spin properties of a large class of materials.*
- The approach here discussed can be generalized and extended to the analysis of a broader class of materials of high interest, including altermagnets.*
- The precise localization of the persistent spin textures for each space group will allow in principle to layout a protocol for designing materials with persistent spin textures near the Fermi level.*
- Finally a remarkable outcome of the research is to show that persistent spin textures are not an exotic property limited to a few special material classes, but actually are universally present in all nonmagnetic solids lacking inversion symmetry.*

I do not have specific criticisms, just a couple of comments.

We are delighted to see that the reviewer finds our paper relevant, timely and well-written, and we appreciate the feedback that helped us to further improve the manuscript.

- It is a bit weird to find the main result on the investigation in the supplementary material, however I understand the need to put Table I in there. I only recommend the authors to emphasize more the presence of the full analysis in the supplementary material Table I and to make reference to the table when discussing the symmetry-based results.

We agree with the referee. We have added a paragraph in the main text discussing Table S1 and have referenced it in the context of our symmetry-based results. Additionally, to improve readability and highlight the relevance of the full table, we have included a condensed version containing a selection of representative space groups directly in the main text.

- I think that the introduction would benefit from a wider introduction of the topic. Although clear to specialists, I think that explaining more the relevance of persistent spin texture and the

implication for material properties would help the audience of material scientists to better appreciate the relevance of the results.

The introduction has been modified to better suit the broader audience of material scientists.

In conclusion, I have minor suggestions to improve the clarity of the manuscript and I recommend the manuscript for publication after minor revision from the authors.

We thank the reviewer once again for the positive comments.

Response to Reviewer 3

This study presents a new theoretical framework for identifying symmetry-protected persistent spin textures (SP-PSTs) in nonmagnetic solids. By systematically analyzing all 230 crystallographic space groups (SGs) using irreducible corepresentations of double grey groups, the authors demonstrate that SP-PSTs are a near-universal property of noncentrosymmetric crystals. The work classifies SP-PSTs into two types (degenerate and nondegenerate bands), identifies their locations in the Brillouin zone (BZ), and validates predictions via first-principles calculations on representative materials (e.g., Be_5Pt , Ag_2Se). The findings challenge the conventional view that PSTs are exotic phenomena limited to specific material classes, offering a transformative approach for designing spintronic materials with robust spin lifetimes. I suggest to publish this paper in NC after the authors address the following comments:

We thank the reviewer for the positive recommendation of our manuscript.

(1) Ambiguity in Universality Claims

The abstract states that PSTs are "universally present in all nonmagnetic solids lacking inversion symmetry." However, the main text clarifies that PSTs are absent in the trivial space group P1 (see Sec. 4 and Supplementary Table S1). This inconsistency risks overstating the universality of PSTs. Recommendation: Revise the abstract to explicitly exclude SG P1, e.g.,

"We demonstrate that PSTs are universally present in all noncentrosymmetric nonmagnetic crystals except those in the trivial space group P1."

We agree with the reviewer. Space group P1 does not have any nontrivial symmetries and it cannot enforce PST in any region. We have now included this statement in the abstract.

(2) Rationale for Using Grey Groups Over Double Groups

The authors employ irreducible corepresentations of double magnetic space groups of type II (grey groups) rather than conventional double groups. While this choice enables the classification of SP-PSTs under time-reversal symmetry (TRS), the rationale remains unclear.

Why do grey groups (which unify space group operations and TRS) provide a more robust framework for identifying type-II PSTs (e.g., in degenerate bands) compared to double groups

(which focus on spin-1/2 states without TRS)?

Does the exclusion of TRS in double groups inherently fail to capture type-II PSTs? If so, explain how grey groups resolve this limitation (e.g., via eigenvalue constraints in Eq. 3).

Recommendation: Expand the Methods section to justify the grey group framework, emphasizing its necessity for predicting PSTs in systems with nonsymmorphic symmetries (e.g., glide mirrors) or TRS-enforced degeneracies.

We thank the reviewer for this question. The rationale for using type-II double magnetic space (grey) groups, rather than conventional double groups, indeed warrants further clarification. As discussed in our response to Reviewer 1 (Comment 1), we adopted double magnetic space groups as the most general ones accounting at the same level of group/representation theory for:

- 1) spin-dependent terms in the Hamiltonian and symmetry properties of half-integer spins under spatial symmetry operations (double groups),
- 2) time-reversal symmetry (magnetic groups),
- 3) spatial symmetries (space groups, including translations, pure point-group symmetry elements as well as screw/glide elements).

Since we are interested in nonmagnetic systems, where T is a symmetry element on its own, we can restrict our analysis to (double magnetic space) grey groups.

In the revised version, we have explained the rationale for using double grey groups and we have added a more detailed discussion in the context of type-II PST examples.